https://doi.org/10.1038/s42003-020-01217-4　　**OPEN**

# Apple latent spherical virus structure with stable capsid frame supports quasi-stable protrusions expediting genome release

Hisashi Naitow [1,5], Tasuku Hamaguchi [1,5], Saori Maki-Yonekura [1], Masamichi Isogai [2], Nobuyuki Yoshikawa [3] & Koji Yonekura [1,4 ✉]

Picorna-like plant viruses are non-enveloped RNA spherical viruses of ~30 nm. Part of the survival of these viruses depends on their capsid being stable enough to harbour the viral genome and yet malleable enough to allow its release. However, molecular mechanisms remain obscure. Here, we report a structure of a picorna-like plant virus, apple latent spherical virus, at 2.87 Å resolution by single-particle cryo-electron microscopy (cryo-EM) with a cold-field emission beam. The cryo-EM map reveals a unique structure composed of three capsid proteins Vp25, Vp20, and Vp24. Strikingly Vp25 has a long N-terminal extension, which substantially stabilises the capsid frame of Vp25 and Vp20 subunits. Cryo-EM images also resolve RNA genome leaking from a pentameric protrusion of Vp24 subunits. The structures and observations suggest that genome release occurs through occasional opening of the Vp24 subunits, possibly suppressed to a low frequency by the rigid frame of the other subunits.

[1] Biostructural Mechanism Laboratory, RIKEN SPring-8 Center, 1-1-1 Kouto, Sayo, Hyogo 679-5148, Japan. [2] Plant Pathology Laboratory, Faculty of Agriculture, Iwate University, Ueda 3-chome 18-8, Morioka, Iwate 020-8550, Japan. [3] Agri-Innovation Center, Iwate University, Ueda 3-chome 18-8, Morioka, Iwate 020-8550, Japan. [4] Advanced Electron Microscope Development Unit, RIKEN-JEOL Collaboration Center, RIKEN Baton Zone Program, 1-1-1 Kouto, Sayo, Hyogo 679-5148, Japan. [5] These authors contributed equally: Hisashi Naitow, Tasuku Hamaguchi. ✉email: yone@spring8.or.jp

Plant viruses invade host cells through a surface wound in the cell wall, often expedited by transmitting insects, and promptly release their genome for translation of their own proteins. Newly proliferating viruses spread to neighboring cells through so-called plasmodesmata, which are channels traversing the cell walls normally for the transport of chemicals and even small proteins between the cells (e.g., refs. [1,2]). This tubular structure is modified by a viral protein called movement protein (MP[2]) to facilitate the entry of viral replication complexes (e.g., refs. [3,4]) or virus particles themselves (e.g., refs. [5–9]). Then, the viral genome is released again, bound for new destinations. In most cases, no specific host receptors are thought to be involved in infection and proliferation. Thus, capsid design for the survival of these viruses should require some basic functions, such as retention of the viral genome inside the virion, genome release, and then its encapsidation with newly synthesized capsid proteins. In addition, the capsid protein of viruses that form a virion during transport through the modified plasmodesma needs to be able to interact appropriately with MP[7,8].

Picorna-like plant viruses are one such group of plant-infecting viruses. They are classified into a large virus order, *Picornavirales*, members of which are nonenveloped RNA spherical viruses of ~30 nm, and which infect various hosts, such as vertebrates, insects, algae, and plants[10]. Although extensive studies have been carried out on these simple plant viruses, key molecular mechanisms still remain unclear. In particular, genome release is a basic and essential step for virus proliferation, and yet little is known about how this occurs.

Apple-latent spherical virus (ALSV) is a picorna-like plant virus found in apple leaves[11]. No symptoms appear upon infection of the virus[12], a phenomenon referred to as "latent infection". Apple trees may be the only naturally infected host as no transmitting insects have been found[13]. Yet, in vivo experiments show that ALSV has a relatively broad spectrum of host plants (*Caryophyllaceae*, *Chenopodiaceae*, *Cryptomeria*, *Fabaceae*, *Cucurbitaceae*, *Gentianaceae*, *Pinus*, *Rosaceae*, *Rutaceae*, *Solanaceae*, and *Arabidopsis*)[14]. Because of this, biotechnological applications have been explored, such as virus-induced gene silencing[15,16], virus-induced flowering[17–19], epitope presentation on the surface of virus particles[19,20], and development of plant vaccines[21–23].

Such picorna-like plant viruses are classified in the family *Secoviridae*. This family contains approximately eight genera, and only a few virion structures in the family have been reported, i.e., cowpea mosaic virus (CPMV[24]), broad bean stain virus (BBSV[25]), bean pod mottle virus (BPMV[26]), and red clover mottle virus (RCMV[27]) in the genus *Comovirus* and tobacco ringspot virus (TRSV[28]) and grapevine fanleaf virus (GFLV[29]) in the genus *Nepovirus*. The virions of the genus *Comovirus* are composed of two proteins, and those of *Nepovirus* only a single protein. The virions of ALSV consist of three proteins, Vp25, Vp20, and Vp24, and they show only a weak sequence homology to other virus capsid proteins. In fact, ALSV was initially classified separately in the family *Sequiviridae*[30], but is now placed in the genus *Cheravirus*, family *Secoviridae*, after the latter was formed from an amalgamation of families *Sequiviridae* and *Comoviridae*.

We have determined the structure of ALSV at 2.87-Å resolution by single-particle cryo-EM with a cold-field emission beam, which allowed de novo modeling of atomic coordinates on the cryo-EM map from a relatively smaller number of particle images with an improved B factor for map sharpening. The structure reveals that the three capsid proteins have characteristic jellyroll folds that contribute to capsid stabilization. Uniquely among the known structures in this virus family, a long N-terminal extension of Vp25 interacts with all other subunits, including a neighboring Vp25 through the inner surface of the capsid, as well

as the genomic RNA. Image analysis resolves RNA leaking from protrusions of the virus particles. Our results suggest how capsid stabilization is achieved, and we propose a mechanism for genome release.

## Results

**Structure analysis of ALSV.** ALSV consists of three protein subunits, Vp25, Vp20, and Vp24, which are coded in the viral genome in this order and named for their approximate molecular weights. We applied single-particle cryo-EM to frozen–hydrated ALSV particles containing genome RNA, isolated from infected plant leaf tissue. A 3D map was reconstructed to 2.87-Å resolution from dose-fractionated movies using RELION-3[31] (Table 1, Supplementary Fig. 1, and Supplementary Movie 1). The resolution is better, and the absolute value of the estimated B factor for map sharpening is low when compared with other viruses in the family (Table 2). This should reflect that higher-resolution signals are well retained, thanks to the cold-field emission beam[32].

The cryo-EM map (Supplementary Fig. 1) revealed side-chain densities, and allowed de novo modeling for the whole virus capsid, except for three short segments (41–55, 100–116, and 146–151) in Vp24 and several residues in the N- and C-termini (Supplementary Fig. 2). After real-space refinement, the final model has good geometry and a high correlation coefficient with respect to the cryo-EM map (Table 1).

**Table 1 Cryo-EM data collection, refinement, and validation statistics.**

| | ALSV (EMD-30375) (PDB 7CHK) |
|---|---|
| *Data collection and processing* | |
| Microscope | CRYO ARM 300 |
| Detector | K2 summit (superresolution mode) |
| Magnification (nominal) | 40,000 |
| Voltage (kV) | 300 |
| Electron exposure (e⁻/Å²) | ~51 |
| Dose rate (e⁻/Å²/s) | ~8.5 |
| Defocus range (μm) | −0.5 ~ −2.0 |
| Physical pixel size (Å) | 1.24 |
| Symmetry imposed | $I_h$ |
| Initial particle images (no.) | 9285 |
| Final particle images (no.) | 8018 |
| Map resolution (Å) | 2.87 |
| FSC threshold | 0.143 |
| *Refinement* | |
| Initial model used (PDB code) | N/A |
| Model resolution (Å) | 2.98 |
| FSC threshold | 0.5 |
| Map-sharpening B factor (Å²) | −29.7 |
| Model composition | |
| Nonhydrogen atoms | 251,820 |
| Protein residues | 31,680 |
| B factors (Å²) | |
| Protein | 49.42 |
| R.m.s. deviations | |
| Bond lengths (Å) | 0.007 |
| Bond angles (°) | 0.635 |
| Validation | |
| MolProbity score[58] | 2.46 |
| Clashscore | 6.68 |
| Poor rotamers (%) | 6.38 |
| Ramachandran plot | |
| Favored (%) | 92.19 |
| Allowed (%) | 7.81 |
| Disallowed (%) | 0.0 |

**Table 2 Cryo-EM structures of the spherical viruses in the family *Secoviridae*.**

| Virus | Resolution (Å) | Number of virus particles | Estimated B factor (Å$^2$) for map sharpening | Electron source | Accelerating voltage (kV) | Content | Reference |
|---|---|---|---|---|---|---|---|
| CPMV | 3.04 | 4998 | −74.6 | Schottky | 300 | Recombinant empty virus-like particle | Hesketh et al.[59] |
| CPMV | 3.44 | 4331 | −107.6 | Schottky | 300 | Including RNA1 (6 kDa) | Hesketh et al.[59] |
| CPMV | 3.94 | 10,850 | −144.1 | Schottky | 300 | Including RNA2 (3.5 kDa) | Hesketh et al.[33] |
| CPMV | 4.25 | 4696 | −185.8 | Schottky | 300 | Naturally formed empty capsid | Hesketh et al.[33] |
| BBSV | 3.22 | 17,233 | −101.0 | Schottky | 300 | Including RNA | Lecorre et al.[25] |
| ALSV | 2.87 | 8018 | −29.7 | Cold-field emission | 300 | Including RNA | This work |

**Structure and subunit interactions**. One protomer is made of Vp25, Vp20, and Vp24 subunits (Fig. 1a), and 60 copies of this unit compose one virion (Fig. 1b). The Vp24 subunits form a pentameric protrusion around a fivefold axis, and the protrusion sits on a base of Vp25 and Vp20. All the three proteins contain jellyroll domains composed of all β-strands, except that Vp25 has extra β1 and β2 (Fig. 1a and Supplementary Fig. 2), and the three proteins also comprise short α-helices and loops. The densities for three short segments (residues 41–55, 100–116, and 146–151) in Vp24 are missing (Supplementary Fig. 2) probably due to their high flexibility (Fig. 1a, c). Residues connecting to and from the missing segments are located at the interfaces to the other surrounding subunits (residues 41–55 to Vp25 and neighboring Vp24, 100–116 to Vp20, and 146–151 to Vp25). Another unit of the capsid can also be chosen as a triangle enclosed with yellow lines in Fig. 1c. It may be convenient to use this triangular unit to describe the structure.

One of the striking features is an ~120-Å-long N-terminal extension of Vp25 (Fig. 1 and Supplementary Fig. S2). When the vertex of the Vp24 pentamer is placed "top" and the structure viewed from outside the virion, the N-terminal extension of Vp25 runs under Vp20 in the same triangle and extends to Vp24 (2Vp24 in Fig. 2a) through the main body of Vp25 (2Vp25) of the neighboring triangle on the right side (Fig. 1a, c). The long extension has intimate hydrogen bonds with Vp20 (0Vp20) and the main body of Vp25 (2Vp25), and forms β-sheets with them (between β1 of 0Vp25 and β9 of 2Vp25, and between β2 of 0Vp25 and β8 of 0Vp20, Fig. 2a, b). In contrast, a smaller number of hydrogen bonds are found with Vp24 (Table 3). The numbers of intersubunit hydrogen bonds in each monomer are 17 for Vp25, 12 for Vp20, and 7 for Vp24. We summarize all interactions among the capsid proteins in Table 3 and Fig. 2a.

We then calculated the buried solvent-accessible surface area (SASA) of each Vp25, Vp20, and Vp24 monomer among the surrounding subunits in the virion (Table 3). The buried SASAs are 6029 Å$^2$ for Vp25, 5092 Å$^2$ for Vp20, and 3607 Å$^2$ for Vp24. Thus, Vp25 and Vp20 have rather extensive interactions with their neighbors, and Vp24, which forms the pentameric protrusion, is more loosely held. Vp25 and Vp20 likely form a rigid capsid frame that supports the looser Vp24 protrusions.

**Comparison with other virus structures**. Here we compare the structure of ALSV with those of other picorna-like plant viruses. We take CPMV (PDB ID: 5MS1)[33] as representative of the genus *Comovirus* (Fig. 3b). Structures in the same genus have been reported for BBSV[25], BPMV[26], and RCMV[27], and they are very similar to that of CPMV. The CPMV capsid is composed of two protein subunits, called large (L) and small (S). The L subunit

contains two domains that correspond to Vp25 and Vp20 of ALSV, and the S subunit corresponds to Vp24—the overall structures and folding are similar (Fig. 3). However, sequence alignment by a widely used program Clustal Omega[34] shows relatively lower homology and poorer match of secondary structure elements (Supplementary Fig. 3), indicating a genetically distant relationship. The Dali server with 3D structure fitting[35] yielded a sequence identity of 9.7% between ALSV and CPMV (all amino acid residues of ALSV were included in the denominator for the calculation).

TRSV[28] and grapevine fanleaf virus (GFLV[29]) from the genus *Nepovirus* in the family *Secoviridae* show the same protein fold (Fig. 3c), but here one large capsid protein comprises three domains, analogous to the three separate proteins of ALSV, and sequence homology is also low between TRSV and ALSV (Supplementary Fig. 3). The structure of ALSV is the first solved in the genus *Cheravirus* to our knowledge, and is the only one with three separate capsid proteins. Interactions of the L and S subunits in the genus *Comovirus* and the corresponding domains in the genus *Nepovirus* are also loose as observed in ALSV.

The long N-terminal extension in Vp25 of ALSV is unique among picorna-like viruses, except that some insect[36] and mammalian viruses[37] exhibit different extensions with a two-stranded β-sheet, which do not interact with the pentameric protrusion (Fig. 3d, e). The capsids of the insect and mammalian viruses are composed of four separate protein subunits, three core proteins (Fig. 3d, e), and one short segment, but the sequence homology is again low (~12.9% between ALSV and insect Triatoma virus (TrV, PDB ID: 3NAP)[36] and ~7.7% between ALSV and mammalian Hepatitis A virus (HAV, PDB ID: 4QPI)[37], obtained in the same way as above using the Dali server for alignment of 3D structures[35].

**Retention and release of genome**. One ALSV virion contains one longer or two shorter single-strand RNA chains. The former (RNA1) encodes proteins for replication, such as helicase, protease, and RNA polymerase, and the latter two (RNA2s) encode a MP and the capsid proteins[30]. There are concentric shells of density inside the capsid and humps of density beneath the Vp24 pentameric protrusions (Fig. 2c). The density corresponds to a symmetrized average of the RNA genome, and this feature is common in single-particle reconstructions of other picorna-like plant viruses (e.g., ref. [24]). The original map showed no clear interaction of capsid protein density with the genome density. However, after applying the autosharpening algorithm implemented in the Phenix suite[38] to the map (Fig. 2c, d), we found that His 113 of Vp20 and Trp 9 of the Vp25 N-terminal extension are involved in a clear interaction with the RNA inside the capsid

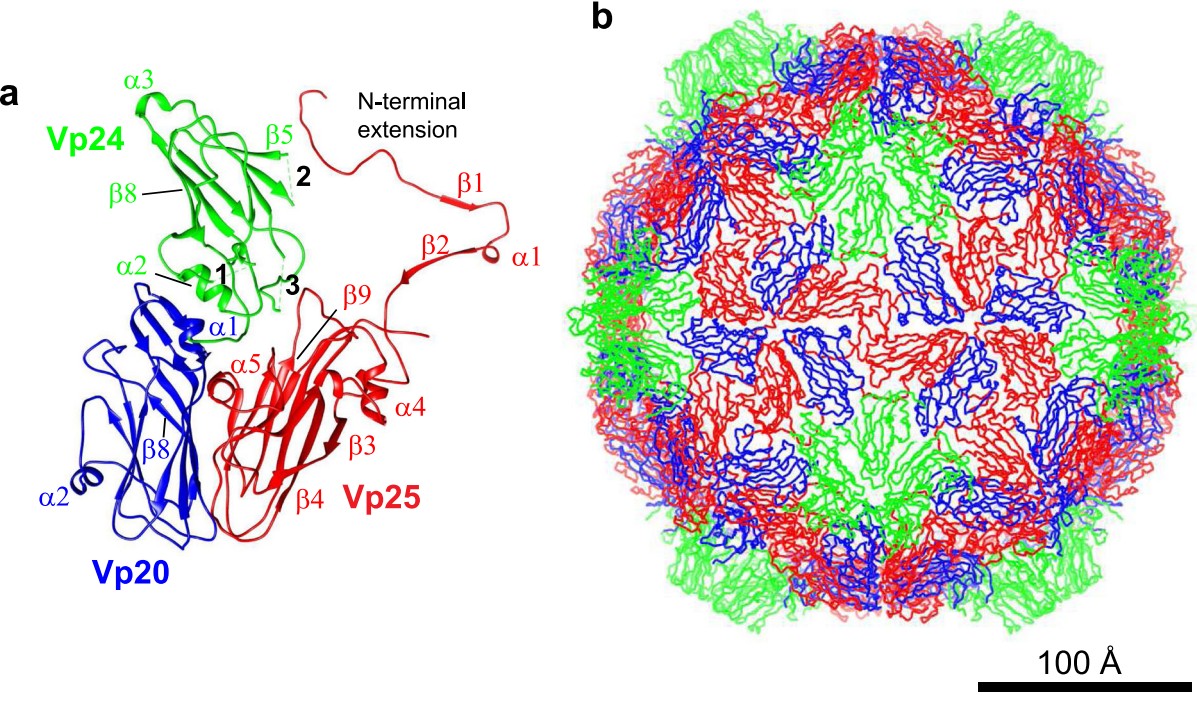

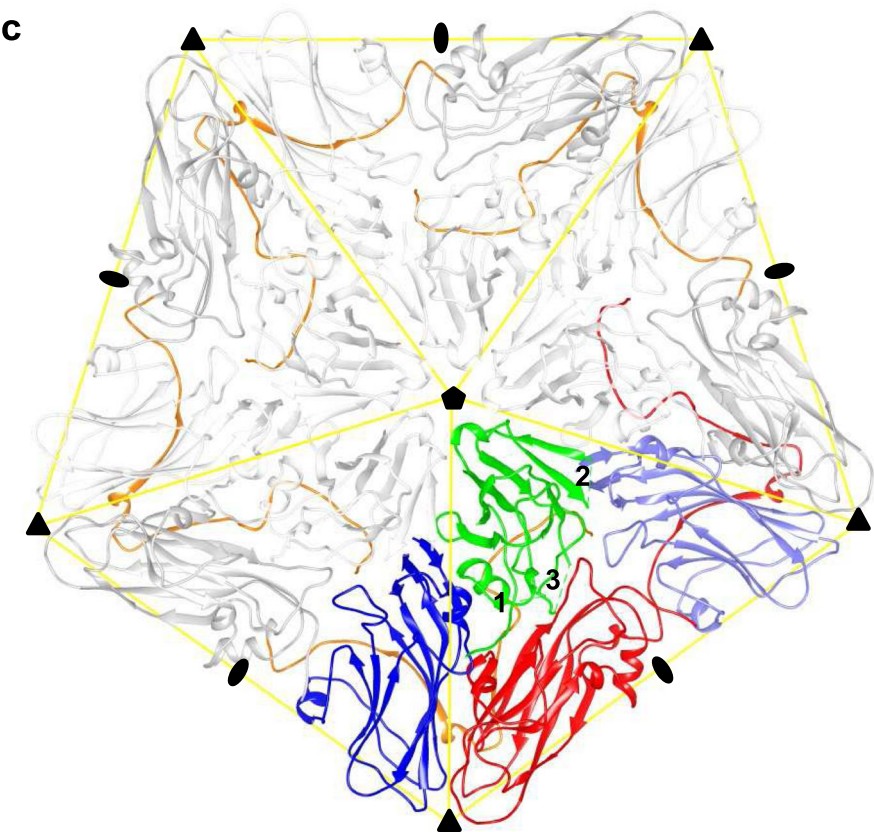

**Fig. 1 Structure of apple-latent spherical virus (ALSV) in the ribbon diagram, determined by single-particle cryo-EM. a** The protomer structure, which is composed of three protein subunits, Vp25 in red, Vp20 in blue, and Vp24 in green, disposed as in other picorna-like plant viruses. Secondary structure elements refer to Supplementary Fig. 2. **b** Overall structure, which comprises 60 copies of the protomer. **c** Five protomers around the fivefold axis. Subunits in one protomer are colored as in (**a**) and others in gray, except for the N-terminal extensions of Vp25 subunits in orange and one Vp20 inside a yellow triangle in light blue. The N-terminal extensions appear to encircle the fivefold axis. The icosahedral two-, three-, and fivefold axes are marked with an ellipse, triangle, and pentamer, respectively. Dashed lines represent three missing segments (41–55, 100–116, and 146–151) in Vp24 and numbered one for residues 41–55, two for 100–116, and three for 146–151 in (**a**) and (**c**).

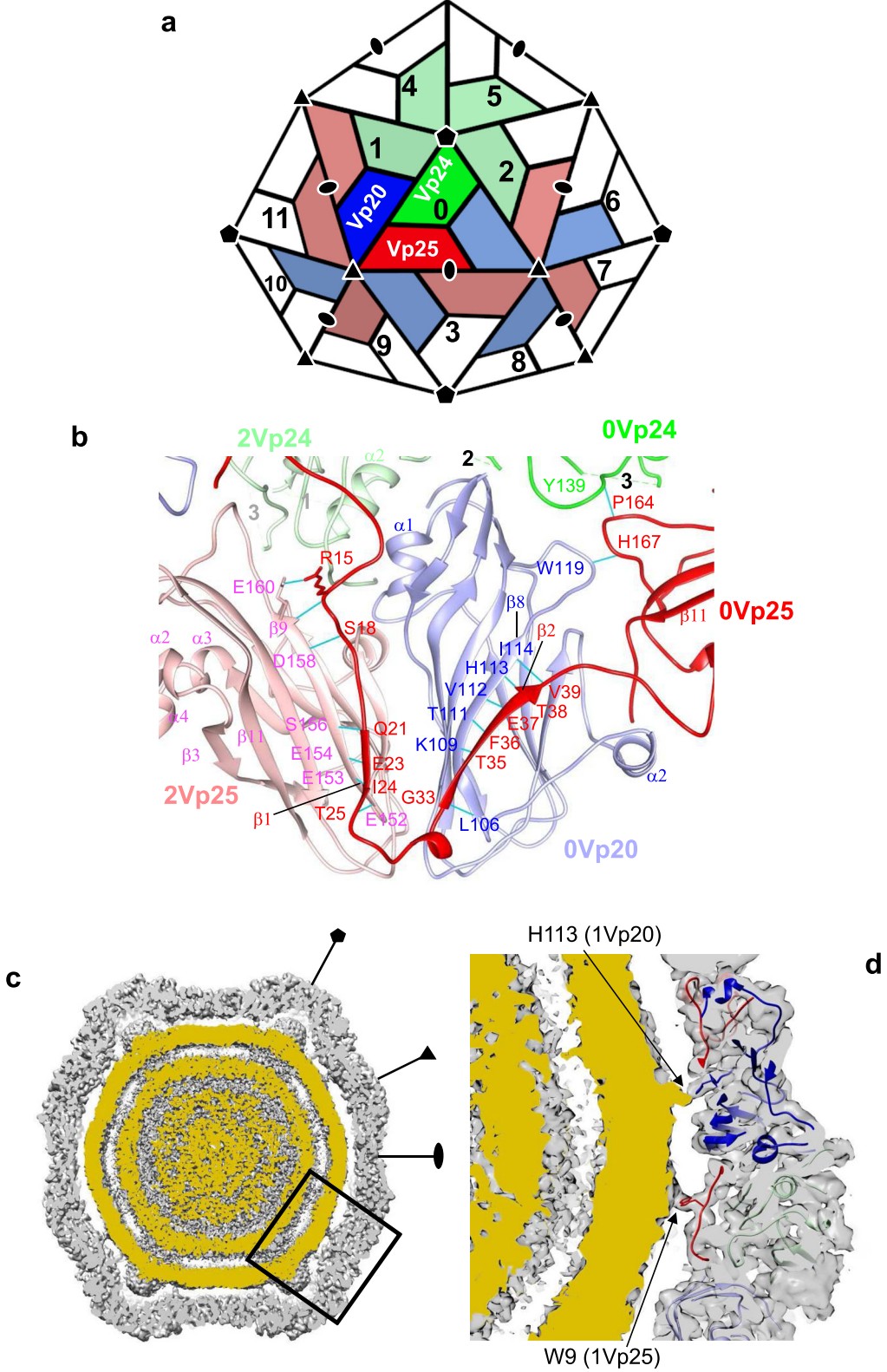

(Fig. 2d). These residues are at different positions from those found in the L subunit in the genus *Comovirus*[25,33]. It is not clear whether this difference represents unique sites for RNA binding in the genus *Cheravirus*, or rather arises just from the enforced symmetry of the RNA density. Nonetheless, the N-terminal extension of Vp25 extensively covers the inner surface of the capsid shell, and likely interacts with the internal RNA genome.

Cryo-EM images of ALSV sometimes contained string-like densities around virus particles as in Fig. 4a and Supplementary Fig. 4a, and they increased after the sample solution was frozen–thawn several times and/or incubated for several days. These strings clearly leaked from the virus particles as some particles appeared partly empty in approximate proportion to the volume of the surrounding strings extruded (Fig. 4b). The strings

**Fig. 2 Interaction of apple-latent spherical virus (ALSV) subunits. a** Arrangements of Vp25, Vp20, and Vp24 subunits in the ALSV virion. Subunits in the protomer are shown in the same color scheme as in Fig. 1, but in deeper colors. Subunits that have contacts with the three in the center triangle (position 0) are also colored and those with no contact are in white. See also Table 3 for buried solvent-accessible surface area (SASA) and numbers of hydrogen bonds between the center subunits and their partners. **b** Interface between Vp25 and Vp20 subunits viewed from inside the capsid. Ribbon diagram showing the secondary structures. Hydrogen bonds in cyan lines are formed between O and NH in peptide bonds, except for Arg 15 of 0Vp25 and Glu 160 of 2Vp25 displayed in stick representation. Dashed lines represent the missing segments in Vp24 and numbered 1, 2, and 3 in black and gray as in Fig. 1a and c. **c**, **d** Interaction between the capsid and internal RNA genome. Cross-sectional views of a sharpened ALSV map by phenix.auto_sharpen[38]. **c** Whole virus showing concentric shells of symmetrized averaged density of RNA genome (shown in ocher) inside the capsid and protrusions beneath the Vp24 pentamer. **d** Zoom-up view of the area enclosed by a black rectangle in (**c**). His 113 of Vp20 and Trp 9 of Vp25 appear to be in contact with the RNA genome. The viewing direction is slightly adjusted from that in (**c**) for clarity. The map is contoured at 3.3 δ.

**Table 3 Interactions of ALSV capsid proteins.**

| Interaction partner[a] | | Buried SASA[b] | Number of hydrogen bonds | Interaction partner[a] | | Buried SASA[b] | Number of hydrogen bonds |
|---|---|---|---|---|---|---|---|
| 0Vp25 | 0Vp20 | 1786 | 7 | 0Vp20 | 0Vp25 | 1771 | 7 |
| | 0Vp24 | 591 | 1 | | 0Vp24 | 443 | 0 |
| | 1Vp20 | 1032 | 3 | | 2Vp24 | 872 | 2 |
| | 1Vp24 | 0 | 0 | | 2Vp25 | 1032 | 3 |
| | 2Vp24 | 367 | 0 | | 3Vp20 | 0 | 0 |
| | 2Vp25 | 829 | 6 | | 3Vp25 | 890 | 0 |
| | 3Vp20 | 889 | 0 | | 6Vp20 | 39 | 0 |
| | 3Vp25 | 535 | 0 | | 7Vp25 | 6 | 0 |
| | 9Vp25 | 0 | 0 | | 8Vp20 | 39 | 0 |
| | 10Vp20 | 0 | 0 | | | | |
| | 11Vp25 | 0 | 0 | | | | |
| Sum | | 6029 | 17 | Sum | | 5092 | 12 |
| 0Vp24 | 0Vp20 | 443 | 0 | | | | |
| | 0Vp25 | 582 | 1 | | | | |
| | 1Vp20 | 871 | 2 | | | | |
| | 1Vp24 | 667 | 2 | | | | |
| | 1Vp25 | 355 | 0 | | | | |
| | 2Vp24 | 684 | 2 | | | | |
| | 2Vp25 | 0 | 0 | | | | |
| | 4Vp24 | 0 | 0 | | | | |
| | 5Vp24 | 5 | 0 | | | | |
| Sum | | 3607 | 7 | | | | |

[a]Refers to Fig. 2a for subunit numbers and positions.
[b]Buried solvent-accessible surface area (SASA).

must correspond to viral RNA genome. Leaking genomes have been reported previously[39,40]. We cut out images that included RNA and part of the virion, and applied 2D classification. Thereby, the 2D-class averages resolved that RNA chains extrude from one of the vertices of the virion, which likely corresponds to the tip of a pentameric protrusion (Fig. 4c). Also, there is a hump or protuberance of genome density beneath each pentameric protrusion of the capsid (Fig. 2c) and a pore with a diameter of ~4 Å there (Fig. 4d). No other pores or clefts with around this size or larger than this are found on the capsid surface. These observations suggest that the genome escapes from the virion at one of the pentameric protrusion sites. Of course, the pore is too small for the genome to pass through and needs to be larger, but the pore would be the only candidate where such opening could occur.

Previous studies by cryo-electron tomography of a poliovirus, which is a human picorna-like virus, suggest that genome release occurs at a deformed part of the capsid around the twofold axis[40]. However, this is unlikely in ALSV, as there are tight interactions of the capsid base composed of Vp25 and Vp20 around the twofold axis (Table 3), and the 2D-class averages in Fig. 4c point to release being at the tip of the pentameric protrusion. Of course, the genome-release mechanism may be different between plant and human-infecting viruses, as the mechanisms of infection and spread are different. Nonetheless, there are indications in other viruses that the pentameric protrusion is related to genome release: the structure of an insect picorna-like virus, honeybee-deformed wing virus, reveals that a RNA segment is bound inside the pentameric protrusion[41]; cryo-EM analysis showed that a tailless bacteriophage ΦX174, which is a spherical virus in another family, is fused to the membrane with a pentameric protrusion of the virion, and the genome is thought to be injected into the host cell through this contact point[42].

## Discussion

The capsid structure requires two apparently contradictory characteristics: it needs to be rigid enough to hold the internal genome, but flexible enough to release the genome to the host cell. The ALSV structure shows a solid base composed of five pairs of Vp25 and Vp20 proteins, and a turret of five Vp24 proteins on top (Fig. 1c). The surface of the virion is tightly patched with this pentagonal unit, except for holes of ~4 Å at the center of five Vp24 subunits (Fig. 4d). The three subunits are structurally supported by the long N-terminal extension of Vp25. This extension makes many intimate hydrogen bonds with Vp20 and neighboring Vp25, but only a few with Vp24 (Fig. 2b; Table 3). The buried

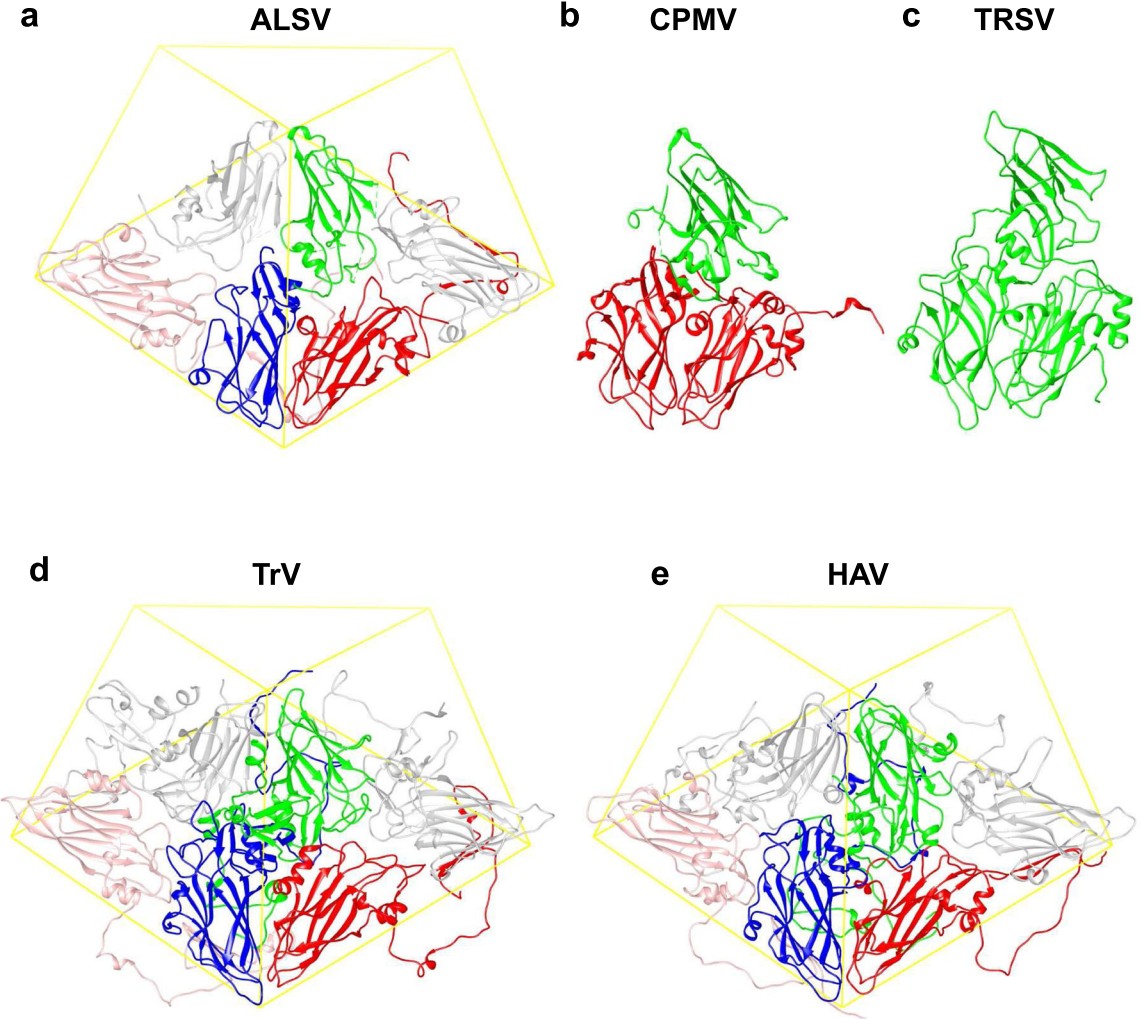

**Fig. 3 Structure comparison of picorna-like plant viruses, an insect and a mammalian picorna viruses. a** Apple-latent spherical virus (ALSV) in the genus *Cheravirus*. Two protomers are shown with the viral lattice and colored as in Fig. 1c, except for the left Vp25 subunit in light red and the right Vp20 in gray. **b** Cowpea mosaic virus (CPMV) (PDB ID: 5MS1)[33] in the genus *Comovirus*. Colored in red for the L and in green for the S subunits. **c** Tobacco ringspot virus (TRSV) (PDB ID: 1A6C)[28] in the genus *Nepovirus* composed of the single-capsid protein colored in green. The sequence identity is 9.7% between ALSV and CPMV, and 8.5% between ALSV and TRSV. The Dali server[35] was used for the alignment (see text). **d** Insect Triatoma virus (TrV, PDB ID: 3NAP)[36]. **e** Mammalian Hepatitis A virus (HAV, PDB ID: 4QPI)[37]. The capsids of TrV and HAV have four separate protein subunits. Two protomers of three core proteins are shown with viral lattices as in (**a**), and one short segment was not identified in the crystal structures. The structures of CPMV, TRSV, TrV, and HAV are superimposed onto that of the ALSV protomer (green, blue, and red in **a**). The sequence identity is 12.5% between ALSV and TrV, and 7.7% between ALSV and HAV calculated in the same way above.

SASAs show the same feature of a looser interaction of Vp24 with the base (Table 3), and the positioning of flexible segments in Vp24 at the interfaces to Vp25 and Vp20 supports this (Fig. 1). Cryo-EM images and the 2D-class averages suggest that RNA could leak from the pentameric protrusion (Fig. 4c).

Recent reports proposed that geometric defects found in spherical viruses by single-particle analysis without symmetry enforcement might be a site for infection and genome release. However, 2D-class averages of ALSV images (Supplementary Fig. 4b) and a 3D reconstruction assuming no icosahedral symmetry do not indicate such defects (Supplementary Fig. 4c). Here we picked up more particles from different datasets and processed together for this test (see Image analysis without icosahedral symmetry in "Methods"). Relatively large defects (e.g., in Fig. 1 of Wang et al.[43]) are unlikely as genome release would be too facile, and the virion would not be suitable for carrying the genome to nearby cells. Then, the defect should only appear at a single defined site rather than at random sites or solid parts of the capsid. The pentameric protrusion in the capsid of ALSV appears eminently suitable to be a release site.

Many ALSV viruses are produced in a plant cell and spread to neighboring cells through a plasmodesma modified with MP[5]. The cryo-EM images (Fig. 4a) suggest that there is a certain low probability that particles lose the RNA genome from the pentameric protrusions (Figs. 2c and 4c; Table 3). The protrusions may make a hinge motion to open a channel to release the genome due to less intimate interactions among themselves and flexible segments of Vp24 at the subunit interface (Figs. 1, 2a, b, and 4d; Table 3). The turret of five Vp24 subunits seems an unstable structure compared with the solid circular base of Vp25 and Vp20 subunits strongly linked through the interfaces and the N-terminal extension. The strongest link of 0Vp24 with a neighbor, judging from the number of hydrogen bonds and buried surface area, is with 1Vp20 of the juxtaposed trimer (Figs. 1, 2a, b; Table 3). This link together with an interaction with the N-terminal extension of 1Vp25 may serve as an anchor for a hinge-

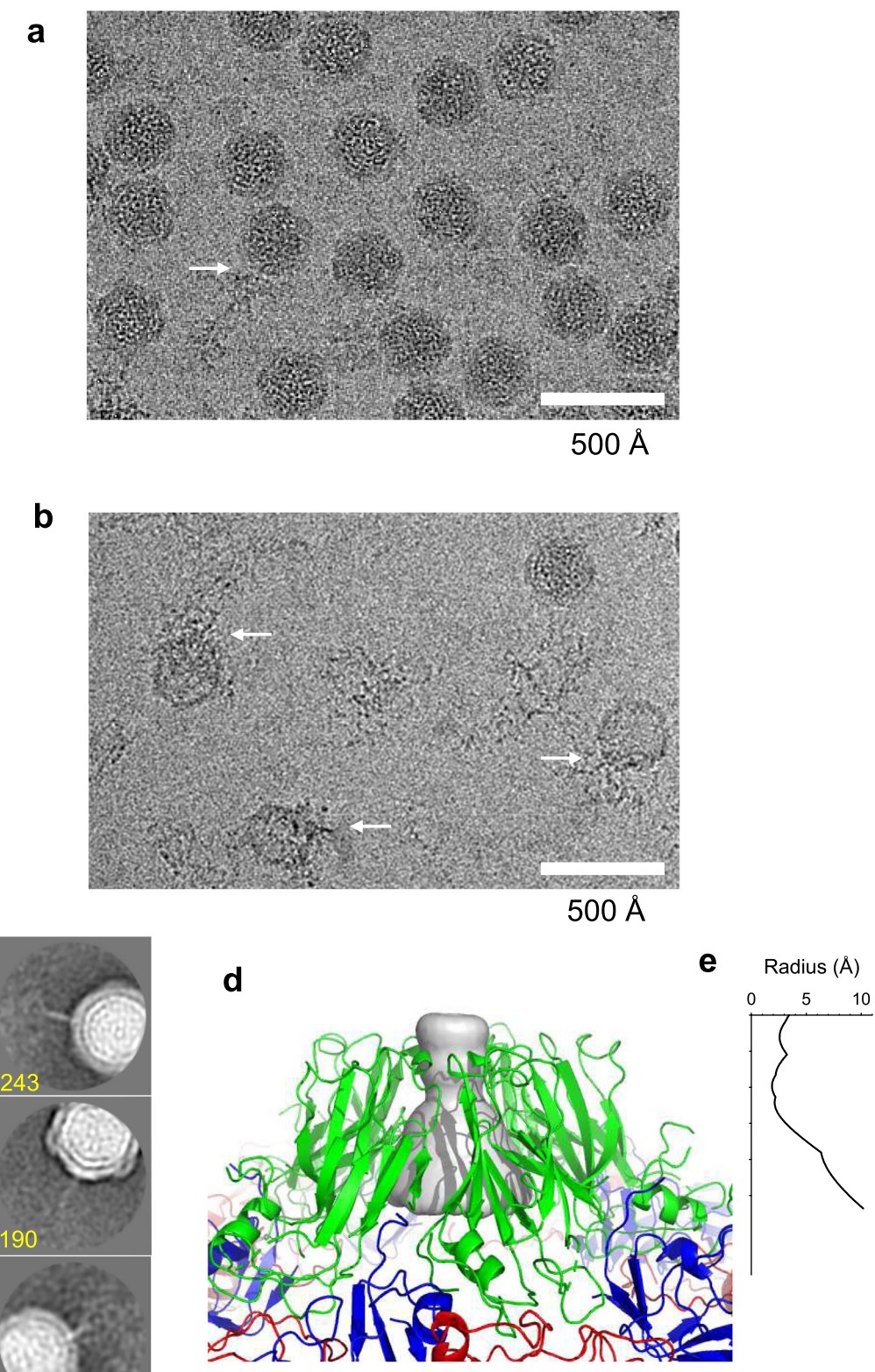

bending motion of Vp24 along the relatively weak 0Vp25 and 0Vp20 interfaces. A mechanism whereby all five Vp24s of a protrusion open like the petals of a flower seems feasible. Making an escape route by dissociation of some or all five Vp24 subunits may also be possible for ALSV and viruses of the *Comovirus* genus since they have separate proteins. The rate of opening of the Vp24 subunits and genome release may depend on RNA

interactions with the Vp25 extension, or may occur spontaneously at a certain low level. Replication of the virus seems to proceed with the leaked RNA from a limited number of viruses.

Our cryo-EM analysis of ALSV reveals the first structure of a picorna-like plant virus in the genus *Cheravirus* to our knowledge. The structure consists of three capsid proteins Vp25, Vp24, and Vp20, arranged so that a rigid frame of Vp25 and

**Fig. 4 Cryo-EM micrographs of apple-latent spherical virus (ALSV) embedded in vitreous ice, and volume rendering of a pore at the center of five Vp24 subunits. a** A typical image used for single-particle reconstruction at 2.87-Å resolution. **b** Images of RNA leakage from many virus particles. Vitrified sample solution was frozen–thawn several times and incubated for several days. Arrows indicate string-like density of leaked RNA. Densities inside viruses appear to be lower in (**b**) than those in (**a**) and lowest in the rightmost one in (**b**), as most RNA had leaked out. **c** Top three of 2D-class averages. RNA leakage is resolved to come from the vertex of the virion. Numbers of particle images that contributed to the corresponding class averages are shown at the bottom. Individual images are shown in Supplementary Fig. 4a. **d** Side view of the pentameric protrusion overlaid with volume rendering inside the central pore in gray. Colored in the same scheme as in Fig. 1 and prepared with PyMol (The PyMOL Molecular Graphics System, Schrödinger, LLC). **e** Plot of the pore radii corresponding to (**d**).

Vp20 subunits is topped by a pentameric protrusion of Vp24 subunits with a center hole of ~4 Å (Figs. 1 and 4d). Image analysis shows the RNA genome leaking from the tip of a protrusion (Fig. 4). The structure of ALSV and observations reported here suggest how the virion is stabilized and points to a possible mechanism for genome release through opening of weakly interacting Vp24 subunits. The N-terminal extension of Vp25 may help in packing of RNA as it is located on the inner surface of the capsid, and does in fact contact the genome (Fig. 2c, d). The structure presented here could, in the future, provide a structural basis to study MP-assisted movement of the picorna-like plant virus through the tubule between cells.

## Methods

**Sample preparation**. ALSV was isolated and purified from infected *Chenopodium quinoa* leaf tissue as described in Li et al.[44]. Purified viruses were suspended in 0.1 M Tris-HCl (pH 7.8), 0.1 M NaCl, and 5 mM MgCl, concentrated to about 10 mg/ml, and kept at −80 °C before use. The sample was negatively stained and checked with a JEM-2100 electron microscope (JEOL, Tokyo). It revealed virus particles of round shape with ~300-Å diameter.

To prepare frozen–hydrated samples, gold was first sputtered on holey carbon film-coated copper grids (Quantifoil R1.2/1.3 or R2/1, Quantifoil Micro Tools GmbH). This reduces beam-induced movement[45], when the samples are exposed to the electron beam. Three microliters of sample solution containing virus particles at a concentration of 1–2 mg/ml were applied onto the grid, blotted manually with filter paper from the reverse side of the grid, and rapidly frozen in liquid ethane with a homemade plunger in a cold chamber with a humidifier.

**Data collection**. The samples were examined at a specimen temperature of ~96 K with a JEOL CRYO ARM 300 electron microscope. A parallel electron beam with a diameter of ~2.3 μm illuminated the sample with a condenser aperture of 150 μm, a spot size of 4, and a convergence angle of 2. Inelastically scattered electrons were removed through an in-column energy filter with an energy slit width of 20 eV. Beam tilt was adjusted to the coma-free axis, and twofold astigmatism was removed by SerialEM[46]. The distribution of ALSV particles in ice appeared to be quite uneven, and most images collected by automated data collection[47] showed no virus particles or very few. Hence, we manually collected images of ALSV particles at a nominal magnification of ×40,000. Dose-fractionated frames were recorded on a Gatan K2 summit in superresolution mode using Gatan DigitalMicrograph. The setting of critical lenses and deflector coils was monitored with ParallEM[32,48]. The physical pixel size of the K2 detector corresponded to 1.24 Å, and one image view included ~100 virus particles, each with a diameter of ~300 Å. Thus, a total of 279 movies were collected. Imaging parameters are shown in Table 1. Magnification was calibrated from gold particles sputtered on carbon, which yields 2.3469- or 2.0325-Å rings.

**Image analysis**. Image processing was performed with RELION-3.0[31]. Image stacks were first drift-corrected, dose-weighted, and summed with the MotionCor2 algorithm[49] implemented in RELION-3. Contrast- transfer function (CTF) parameters were estimated with CTFFIND[50], and particles were manually picked up through the image display of RELION for initial templates. Then, automatic particle picking and reference-free 2D classification were carried out by following the standard protocol of RELION-3[31]. Here, picked particles were edited by visual inspection before the 2D classification. Creation of a de novo initial 3D map, 3D classification based on the initial map, and 3D auto-refinement were all done with icosahedral symmetry enforcement. After applying refinement of defocus and Bayesian polishing for motion correction of individual particles[31,51], the final 3D structure was reconstructed from particle images (8,018) recorded on 110 movie stacks. Beam tilts were also corrected for each micrograph (Supplementary Fig. 1f). Image analysis statistics are shown in Table 1.

A soft mask was estimated and applied to the two half-maps in the post-processing process of RELION. B-factor estimation and map sharpening were also carried out in this step. The resolution was measured to be 2.87 Å based on the gold standard FSC 0.143 criteria (Supplementary Fig. 1e)[52]. The density map clearly reveals the side-chain densities of amino acids (Supplementary Figs. 1b–d), supporting the estimated resolution. Local resolution was calculated using RELION from unfiltered half-maps (Supplementary Fig. 1a). Details related to cryo-EM single-particle analysis are summarized in Table 1.

**De novo model building**. As virus structures in this genus, and even parts thereof, have not been solved before, we carried out de novo model building onto the 3D reconstruction. As the cryo-EM map clearly revealed side-chain densities, an automated procedure implemented in ARP/wARP[53] produced a reasonable initial atomic model. The protomer model was manually adjusted with COOT[54], and then the model of the whole virus was refined by using phenix.real_space_refine[55] with icosahedral symmetrization. This gave good geometries and high cross-correlation (CC) values against the cryo-EM map (Table 1). The structures in Figs. 1, 2b–d, 3, Supplementary Figs. 1a–d and 4c were prepared with UCSF Chimera[56]. Intersubunit-buried SASAs and hydrogen bonds were calculated with UCSF Chimera[56]. The surface pore of the capsid was analyzed by HOLE[57].

The 3D-density map resolved amino acids 2–216 (Vp25), 3–168 (Vp20), and 6–190 (Vp24), and residues 41–55, 100–116, and 146–151 in Vp24 are missing on the map.

**Image analysis of the genome**. The reconstructed volume of ALSV was sharpened by phenix.auto_sharpen[38] for visualization of finer structures. This revealed residues contributing to interactions with the internal densities of the symmetrized average RNA genome (Fig. 2c, d).

Images, including leaked RNA and part of the virion, were cut out (Supplementary Fig. 4a) from motion-corrected micrographs of frozen–hydrated virus after the sample solution was frozen–thawn several times and/or incubated for several days. Then, a total of 1142 cut-out images were subjected to 2D classification with an option of "Ignore CTFs until first peak". The top three class averages revealing leaked RNA (Fig. 4c) include 243, 190, and 90 particle images.

**Image analysis without icosahedral symmetry**. We picked up more ALSV particles from images taken with automated data collection software JADAS[47], although the yield rate was not good due to the uneven distribution of viruses in ice as above. Particle images were processed together with manually collected ones, and 2D classification was applied to a total of 37,328 particles from 1,296 movie stacks with an option of "Ignore CTFs until first peak," the setting of which is expected to be suitable for detection of finer features. A 3D map was reconstructed without icosahedral symmetry from a total of 28,633 particles. Icosahedral symmetrization of this combined dataset did not improve the resolution of the structure probably due to an increased heterogeneity among the datasets.

**Statistics and reproducibility**. The parameters and statistics of data collection, image analysis, and de novo atomic model building are shown in Table 1.

**Reporting summary**. Further information on research design is available in the Nature Research Reporting Summary linked to this article.

## Data availability

A cryo-EM-density map of ALSV and its atomic coordinates have been deposited in EMDataResource under the accession number EMD-30375 and in the Protein Data Bank under the accession number 7CHK, respectively. Sequence data of ALSV RNA1 and RNA2 are accessible in the DNA Data Bank of Japan under accession numbers AB030940 and AB030941, respectively. Source data for the graph in Fig. 4e are available as Supplementary Data.

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

## Acknowledgements

We thank D.B. McIntosh for help in improving the paper. This work was partly supported by Japan Society for the Promotion of Science Grant-in-Aid for Scientific Research Grant 16H04757 (to K.Y.), Japan Society for the Promotion of Science Grant-in-Aid for Challenging Exploratory Research Grant 24657111 (to K.Y.), the RIKEN Pioneering Project, Dynamic Structural Biology (to K.Y.), and the Cyclic Innovation for Clinical Empowerment (CiCLE) from the Japan Agency for Medical Research and Development, AMED (to K.Y.).

## Author contributions

H.N., T.H., S.M.-Y., M.I., N.Y., and K.Y. designed the experiments; M.I. prepared the sample; H.N., T.H., S.M.-Y., and K.Y. collected image data; H.N., T.H., S.M.-Y., and K.Y. analyzed the data; T.H. reconstructed the cryo-EM map; H.N. built the atomic model; N.H., H.T., N.Y., and K.Y. wrote the paper.

## Competing interests

The authors declare no competing interests.
