## [Peer Review File · Communications Biology]

Reviewers' comments:

Reviewer #1 (Remarks to the Author):

The manuscript titled " Structure of apple latent spherical virus points to a stable capsid frame supporting quasi-stable protrusions expediting genome release " by Hisashi Naitow et al present a 2.7 angstrom resolution structure of a picorna-like plant virus, apple latent spherical virus (ALSV), which closely resembles the structures of many picornaviruses. The authors found that the long N-terminal extension of capsid protein VP25 interacts with three neighbored capsid proteins of one Vp20, one Vp24 and one another VP25, which, together with the jellyroll fold of the three capsid proteins, make a major contribution to the capsid stabilization of ALSV. This is somewhat interesting. However, the authors failed to properly present their structural data and to give reasonable interpretations to support the proposed mechanism of viral genome release.

1. Is the map resolution 2.67 or 2.7 angstrom? Please be consistent.
2. The authors mentioned many times that their reconstruction or model of ALSV has better data statistics than others from the same family, but they didn't present the comparison data at all, e.g. "The number of molecular images (8,018; Supplementary Table 1) and the absolute value of the estimated B-factor (-22.2 Å²; Supplementary Table 1) for map sharpening are low if compared with other spherical viruses in a similar resolution range in EMDDataResource" or "Indeed, the reconstruction of ALSV shows better data statistics over other viruses in the family" in Page 5
3. In page #6, "The missing segments are located at the interfaces to the other surrounding subunits...." It is hard to understand how the authors could conclude where the missing segments are.
4. In Fig.2, it is no need to show the hydrogen bonds between β strands.
5. "Other interactions are formed among Vp24 subunits in the pentamer and around a three-fold axis", this sentence is hard to understand, please rephrase it.
6. In page #9, "Nonetheless, the N-terminal extension of Vp25 extensively covers the inner surface of the capsid shell, and interactions with the RNA seem likely." Please rephrase the sentence.
7. In page #9, "Cryo-EM images of ALSV sometimes We cut out images that included leaked RNA and part of the virion, and applied a 2D classification" The description about RNA leaking is redundant and is also hard to follow, please rephrase these sentences.
8. In page 10, "Also, there are a hump or protuberance of genome density No other pores nor clefts with around this size or larger than this are found on the capsid surface". A pore with a diameter of ~ 4Å is not large enough to let viral genome leak.
9. In page 11, "Indeed, cryo-EM images and the 2D class average resolves RNA leaking from the pentameric protrusion". the experimental data is not solid enough to support the authors to draw the conclusion, because they didn't show data to exclude the viral genome could leak out of the viral capsid from other places of the capsid.
10. In page 14, "...done with symmetry enforcement", please indicate the symmetry applied

Reviewer #2 (Remarks to the Author):

In this study, the authors report a structure of apple latent spherical virus (ALSV) at 2.7-Å resolution using single-particle cryo-EM. They found a unique structure composed of three capsid proteins that have not been reported before. Besides, a RNA genome release mechanism was also proposed based on structure analysis. The manuscript was written in good logic, and the results were convincing. I have no major concerns about the manuscript and recommend this work for publication after revision.

Comments

- Overall -

It would be easier for the reviewers to raise comments if the line number was indicated in the manuscript.

- Abstract -

1. The full name of cryo-EM should be pointed out when first mentioned.

- Results and discussion -

2. "molecular images" should be changed to "particle images" or "particles" for the general use.

3. Although the estimated B-factor for map sharpening can reflect the resolution signal to some extent, the "Henderson-Rosenthal plot (also called "B-factor plot")" is a better evaluator. The authors should carefully read the original paper published in 2003 by Henderson and Rosenthal. The author can also check some published papers to understand more on how to generate the plot and show the B-factor for datasets used in this paper.

4. The "better data statistics" used in the same section is ambiguous. I think changing it to "better data quality" is much better.

5. Supplementary Fig. 1b only shows representative map densities, and thus cannot support the sentence "The cryo-EM map reveals most of the side chains as well as the main chain". Please modify this sentence or use a more comprehensive method to analyze the map and model quality and generate a new figure.

6. In the "Structure and subunit interactions" section, change "except for" to "except". And at the same sentence, I think "Supplementary Fig. 1a" instead of "Supplementary Fig. 2" is more appropriate.

7. In the sentence "The long extension has intimate hydrogen bonds with Vp20 and...", Vp25 should be described more precisely.

8. In the "Comparison with other virus structures" section, please reword the long sentence "The long N-terminal extension in Vp25 of ALSV..." for more clear meaning. And there is an obvious grammar error there.

9. The first sentence of the "Retention and release of genome" section is ambiguous, does ALSV contain one longer RNA1 and two shorter RNA2s?

10. What does 24 stand for in "(e.g. 24)"? a reference?

11. After a semicolon, "Cryo-EM" should be changed to "cryo-EM".

12. In the second paragraph of "Mechanisms of genome release" section, the "Supplementary Fig. 4b" and "Supplementary Fig. 4c" should be changed to "Supplementary Fig. 5b" and "Supplementary Fig. 5c", respectively. Besides, Supplementary Fig. 5a should be cited before them for the principle of sequential citation of figures. There is also a missing right parenthesis in this paragraph.

13. Reword the long sentence "Making an escape route by dissociation of some or all five Vp24 subunits, while possible for ALSV..." for clarity. Besides, I think the observation of the genome leaking from the protrusion is still consistent with the mechanism proposed here.

- Conclusion -

14. In the last sentence, please change "though" to "through".

- Materials and Methods -

15. Please delete extra spaces in the unit of "mg / ml".

- Supplementary Figure 3 -

16. Please change one of two "(b)" to "(c)", and indicate how to adjust these structures.

- Table 1 -

17. I'm kind of confused about the virus CPMV in two different papers. Its resolution resolved in 2017 is lower than that in 2015.

- Table 2 -

18. Please give the full name of "SASA" below the table.

- Fig. 2 -

19. "D" should be lower case.

- Supplementary Table 1 -

20. A missing space in the table title.

Kaiming Zhang

Reviewer #3 (Remarks to the Author):

Review of:

Structure of apple latent spherical virus points to a stable capsid frame supporting quasi-stable protrusions expediting genome release

Hisashi Naitow^{1, #}, Tasuku Hamaguchi^{1, #}, Saori Maki-Yonekura¹, Masamichi Isogai², Nobuyuki Yoshikawa³ and Koji Yonekura^{1, 4, *}

Summary:

The authors present a high-resolution capsid structure for apple latent spherical virus by cryo-EM. This represents the first structure of a viral capsid within the Cheravirus genus. The structure is used to

support a model for genome release through an annulus at the 5-fold symmetry axis. This model is supported by inspection of the atomic model and quantification of both hydrogen bonding networks and solvent hidden surface area at the coat protein interfaces. Furthermore, the authors present isolated particles and a 2d class average showing amorphous density extruding from the virion's vertex, which is presumed to be the genome. The data presented are of high quality and the authors achieve a high resolution with a relatively small number of particles.

Comments/recommendations:

1. Throughout the authors use very decisive language '*The structures and observations suggest that genome release **always** occurs through occasional opening of the Vp24 subunits, possibly suppressed to a low frequency by the rigid frame of the other subunits.*'

One may want to relax such language to reflect that the model proposal here is supported only by intimations made from the structure presented, which may not be the entire picture.

2. The authors state that the N-terminal extension of Vp25 is 'Uniquely' interacting with other coat protein protomers and the genome. It will be important to clarify what they mean here, unique amongst all viruses? Unique amongst Cherviruses? Unique amongst the three protomers?
3. The authors use 'similar protein folding' and to describe structures of the same protein fold. This may be confusing to the reader.
4. It might be useful to show the structures of the coat proteins with n N-terminal extensions similar to those of ALSV identified by Dali (5L7O, 4QPI).
5. The authors present a 2d class average of what is presumably RNA extruding from the virion. How many particles are in this 2d class average? It is presumably a very small number given the size of the dataset. This limits the potential of the authors to make a 3d reconstruction of RNA release at the vertex.
6. The authors state that a 3d reconstruction of ALSV with no imposed symmetry does not show any surface defects indicative of genome release. This is unsurprising with such a small dataset. Have the authors tried to collect a larger dataset and classify out particles with defects by 3d classification?

7. The authors need to be sure to reference all relevant software used for image processing and model building/refinement. E.g. COOT for model building.

The conclusions made by the authors are reasonable given the data presented. However, a more thorough examination of genome release could be made with a larger dataset and further analysis. 3d classification of particles with/without RNA extruding and symmetry expansion and focussed classification of particles with/without extruding RNA would both likely give further insight into structural differences in at the vertices during RNA escape. Nonetheless the structure presented gives valuable insight into the structures of viruses within the cheravirus genus.

Responses:

To Reviewer #1

> 1. Is the map resolution 2.67 or 2.7 angstrom? Please be consistent.

Corrected to 2.67 Å throughout the manuscript.

> 2. The authors mentioned many times that their reconstruction or model of ALSV has better data statistics than others from the same family, but they didn't present the comparison data at all, e.g. "The number of molecular images (8,018; Supplementary Table 1) and the absolute value of the estimated B-factor (-22.2 Å²; Supplementary Table 1) for map sharpening are low if compared with other spherical viruses in a similar resolution range in EMDataResource" or "Indeed, the reconstruction of ALSV shows better data statistics over other viruses in the family" in Page 5

We clarify the comparison as,

"The resolution is better and the absolute value of the estimated B-factor for map sharpening is low if compared with other viruses in the family (Table 1)".

> 3. In page #6, "The missing segments are located at the interfaces to the other surrounding subunits...." It is hard to understand how the authors could conclude where the missing segments are.

Corrected as,

"Residues connecting to and from the missing segments are located at ...".

> 4. In Fig.2, it is no need to show the hydrogen bonds between β strands.

We removed the lines of hydrogen bonds between β strands as below, but we think it is unclear which residues contribute to the formation of hydrogen bonds in the new figure. So, we would like to keep Fig. 2b unchanged.

> 5. “Other interactions are formed among Vp24 subunits in the pentamer and around a three-fold axis”, this sentence is hard to understand, please rephrase it.

We deleted this part as we think that this is not an important description, and added a sentence as,

“We summarize all interactions among the capsid proteins in Table 2 and Fig. 2a”.

> 6. In page #9, “Nonetheless, the N-terminal extension of Vp25 extensively covers the inner surface of the capsid shell, and interactions with the RNA seem likely.” Please rephrase the sentence.

Rephrased as,

“Nonetheless, the N-terminal extension of Vp25 extensively covers the inner surface of the capsid shell, and likely interacts with the internal RNA genome”.

> 7. In page #9, “Cryo-EM images of ALSV sometimes We cut out images that included leaked RNA and part of the virion, and applied a 2D classification” The description about RNA leaking is redundant and is also hard to follow, please rephrase these sentences.

Corrected as,

“We cut out images that included RNA and part of the virion, and applied 2D classification”.

> 8. In page 10, “Also, there are a hump or protuberance of genome density No other pores nor clefts with around this size or larger than this are found on the capsid surface”. A pore with a diameter of $\sim 4\text{\AA}$ is not large enough to let viral genome leak.

We agree that the $\sim 4\text{\AA}$ pore is too small for the genome to pass through. We think Vp24 proteins may make a hinge motion to open a channel to release the genome. We described this model in the text and also that the pore would be the only candidate where such opening could occur, as other sites appear to be too solid in the obtained structure. We added this explanation in the text.

> 9. In page 11, “Indeed, cryo-EM images and the 2D class average resolves RNA leaking from the pentameric protrusion”. the experimental data is not solid enough to support the authors to draw the conclusion, because they didn’t show data to exclude the viral genome could leak out of the viral capsid from other places of the capsid.

We agree with the reviewer’s suggestion, but at the same time it is difficult to exclude the possibility that the RNA leaks out from other places of the capsid. Nonetheless, top 2D class averages newly calculated from more increased particle images have revealed RNA leak from the vertex of the virion again in the new Fig. 4c. No RNA leakage was found in any other sites in this analysis. Still, we think a less decisive expression is more appropriate here and modified the text as, “Cryo-EM images and the 2D class averages suggest that RNA could leak from the pentameric protrusion”.

> 10. In page 14, “...done with symmetry enforcement”, please indicate the symmetry applied

Corrected as “... done with icosahedral symmetry enforcement”.

To Reviewer #2

- Overall –

> It would be easier for the reviewers to raise comments if the line number was indicated in the manuscript.

Done.

- Abstract -

> 1. The full name of cryo-EM should be pointed out when first mentioned.

Corrected.

> 2. “molecular images” should be changed to “particle images” or “particles” for the general use.

Corrected.

> 3. Although the estimated B-factor for map sharpening can reflect the resolution signal to some extent, the “Henderson-Rosenthal plot (also called “B-factor plot”)” is a better evaluator. The authors should carefully read the original paper published in 2003 by Henderson and Rosenthal. The author can also check some published papers to understand more on how to generate the plot and show the B-factor for datasets used in this paper.

Thank you for suggestion. We know that the Henderson-Rosenthal plot is well accepted in the community and the plot can be used to estimate the number of particle images to be expected for a target resolution and to calculate the Henderson-Rosenthal B-factor. We do not intend to calculate how many particles are needed to extend the resolution from the reported value in this work. Here we would simply like to compare how much B-factors are estimated for amplification of high-resolution terms in the reported cryo-EM structures in this virus family. There are no Henderson-Rosenthal B-factors in the other papers shown in Table 1, and so direct comparison is difficult.

Nonetheless, we calculated the Henderson-Rosenthal plot as below. The fitted value appears to be not excellent. However, in general, the value of the Henderson-Rosenthal B-factor becomes better with a steeper slope of the plot, and the slope becomes steeper when a poorer resolution is estimated for reconstructions from smaller numbers of particles. Here, the resolution is high even from 100 particle images ($\sim 4 \text{ \AA}$ resolution) thanks to the high symmetry of the icosahedral virus, which likely yields a worse estimate of the Henderson-Rosenthal B-factor. Of course, the plot depends on the sample itself.

Thus, we think that the map sharpening B-factor is more appropriate here.

Particles	LnParticle	Resolution	1/Resolution^2
100	4.60517019	4.04	0.061268503
200	5.29831737	3.65	0.075060987
400	5.99146455	3.55	0.079349335
800	6.68461173	3.22	0.096446896
1600	7.37775891	2.99	0.111855572
3200	8.07090609	2.82	0.125748202
6400	8.76405327	2.72	0.13516436
8018	8.98944429	2.67	0.140274096

> 4. The “better data statistics” used in the same section is ambiguous. I think changing it to “better data quality” is much better.

We removed this sentence according to Reviewer #1’s suggestion.

> 5. Supplementary Fig. 1b only shows representative map densities, and thus cannot support the sentence “The cryo-EM map reveals most of the side chains as well as the main chain”. Please modify this sentence or use a more

comprehensive method to analyze the map and model quality and generate a new figure.

We deleted this expression and modified the text as, “The cryo-EM map (Supplementary Fig. 1) revealed side chain densities, and allowed de novo modelling for the whole virus capsid except for three short segments (41 – 55, 100 – 116 and 146 – 151) in Vp24 and several residues in the N- and C-termini (Supplementary Fig. 2)”.

> 6. In the “Structure and subunit interactions” section, change “except for” to “except”. And at the same sentence, I think “Supplementary Fig. 1a” instead of “Supplementary Fig. 2” is more appropriate.

We rephrased as,
“except Vp25 has extra β 1 and β 2 of Vp25 (Fig. 1a and Supplementary Fig. 2), and the three proteins also comprise short α -helices and loops”. Supplementary Fig. 1a is an overall structure showing local resolutions in multi-color representation according to the scale bar, and we think Fig. 1a is more appropriate here.

> 7. In the sentence “The long extension has intimate hydrogen bonds with Vp20 and...”, Vp25 should be described more precisely.

Corrected as,
“The long extension has intimate hydrogen bonds with Vp20 (0Vp20) and the main body of Vp25 (2Vp25), ... ”.

> 8. In the “Comparison with other virus structures” section, please reword the long sentence “The long N-terminal extension in Vp25 of ALSV...” for more clear meaning. And there is an obvious grammar error there.

Corrected as,
“The long N-terminal extension in Vp25 of ALSV is unique among picorna-like viruses, except some insect³⁶ and mammalian viruses³⁷ exhibit different extensions with a two-stranded β -sheet, which do not interact with the pentameric protrusion (Figs. 3d and e)”.

> 9. The first sentence of the “Retention and release of genome” section is ambiguous, does ALSV contain one longer RNA1 and two shorter RNA2s?

Corrected as,

“One ALSV virion contains one longer or two shorter single-strand RNA chains. The former (RNA1) encodes proteins for replication such as helicase, protease, RNA polymerase etc., and the latter two (RNA2s) encode a MP and the capsid proteins³⁰.”

> 10. What does 24 stand for in “(e.g. 24)” a reference?

Yes. Corrected as “(e.g. reference 24)”.

> 11. After a semicolon, “Cryo-EM” should be changed to “cryo-EM”.

Corrected.

> 12. In the second paragraph of “Mechanisms of genome release” section, the “Supplementary Fig. 4b” and “Supplementary Fig. 4c” should be changed to “Supplementary Fig. 5b” and “Supplementary Fig. 5c”, respectively. Besides, Supplementary Fig. 5a should be cited before them for the principle of sequential citation of figures. There is also a missing right parenthesis in this paragraph.

Following Reviewer #3’s suggestion, we made the new Figs. 3d and e (comparison of the TrV and HAV structures with ALSV) and moved the original Supplementary Fig. 5 to Supplementary Fig. 4. We changed the number of the figures accordingly, and cited “Supplementary Fig. 4a” as,

“Cryo-EM images of ALSV sometimes contained string-like densities around virus particles as in Fig. 4a and Supplementary Fig. 4a, ...” in the 2nd paragraph of “Retention and release of genome” section before Supplementary Figs. 4b and c.

> 13. Reword the long sentence “Making an escape route by dissociation of some or all five Vp24 subunits, while possible for ALSV...” for clarity. Besides, I think the observation of the genome leaking from the protrusion is still consistent with the mechanism proposed here.

We agree with the reviewer’s suggestion and rephrased the sentence as,

“Making an escape route by dissociation of some or all five Vp24 subunits may also be possible for ALSV and viruses of the *Comovirus* genus since they have separate proteins”.

> 14. In the last sentence, please change “though” to “through”.

Corrected.

> 15. Please delete extra spaces in the unit of “mg / ml”.

Deleted the spaces.

- Supplementary Figure 3 -

16. Please change one of two “(b)” to “(c)”, and indicate how to adjust these structures.

The structures of CPMV and TRSV are superimposed onto that of the ALSV protomer (green, blue and red in a) by secondary structure matching (SSM) of COOT. We added this description in the legend of the new Fig. 3.

> - Table 1 -

> 17. I’m kind of confused about the virus CPMV in two different papers. Its resolution resolved in 2017 is lower than that in 2015.

The 2015 paper reported the structures of wild type CPMV containing 6 kDa RNA-1 and of a recombinant empty virus-like particle, while the 2017 paper reported the structures of the wild-type CPMV containing 3.5 kDa RNA-2 and of naturally-formed empty capsids. We described this explanation in Table 1.

> - Table 2 -

> 18. Please give the full name of “SASA” below the table.

Corrected.

> - Fig. 2 -

> 19. “D” should be lower case.

Corrected.

> - Supplementary Table 1 -

> 20. A missing space in the table title.

We do not figure out what the reviewer indicated here.

To Reviewer #3

> 1. Throughout the authors use very decisive language '*The structures and observations suggest that genome release always occurs through occasional opening of the Vp24 subunits, possibly suppressed to a low frequency by the rigid frame of the other subunits.*' One may want to relax such language to reflect that the model proposal here is supported only by intimations made from the structure presented, which may not be the entire picture.

We deleted “always” here and made expressions less decisive throughout the manuscript.

> 2. The authors state that the N-terminal extension of Vp25 is 'Uniquely' interacting with other coat protein protomers and the genome. It will be important to clarify what they mean here, unique amongst all viruses? Unique amongst Cheraviruses? Unique amongst the three protomers?

“Uniquely” means unique among the known structures in this virus family. We corrected as,

“Uniquely among the known structures in this virus family, a long N-terminal extension of Vp25 interacts with all other subunits including a neighboring Vp25 through the inner surface of the capsid, as well as the genomic RNA”.

> 3. The authors use 'similar protein folding' and to describe structures of the same protein fold. This may be confusing to the reader.

Corrected to “the same protein folding”.

> 4. It might be useful to show the structures of the coat proteins with n N-terminal extensions similar to those of ALSV identified by Dali (5L7O, 4QPI).

We are sorry that the PDB ID of the insect Triatoma virus was wrong. The correct ID is 3NAP. We made a new figure (now Figs. 3d and e) to show the comparison of the structures.

> 5. The authors present a 2d class average of what is presumably RNA extruding from the virion. How many particles are in this 2d class average? It is presumably a very small number given the size of the dataset. This limits the potential of the authors to make a 3d reconstruction of RNA release at the vertex.

In the original manuscript, we obtained the 2D class average from 43 out of 114 particle images to make the original Fig. 3c. This was written in Section 4.5 of the original manuscript. In general, we can get a clear 2D average even from such an image number, as the signal-to-noise ratio is proportional to the square root of the image number (N). Indeed, we obtained clear 2D class averages from small numbers of particle images (Please see the new Supplementary Fig. 4b). Nonetheless, we processed ~11 times more particle images and obtained similar class averages to the previous one. Please see the new Fig. 4c. No RNA leakage was found in any other sites in the 2D class averages.

It is difficult to obtain a 3D structure of this site with leaking RNA, as RNA parts are random. We also tried a focused refinement of the pentameric protrusion by masking one pentamer off with ~ 4.7 times more particle images, but failed to obtain any solid structure even for that without RNA leak. This is probably because signals from these small parts are too low compared to the whole virion.

> 6. The authors state that a 3d reconstruction of ALSV with no imposed symmetry does not show any surface defects indicative of genome release. This is unsurprising with such a small dataset. Have the authors tried to collect a larger dataset and classify out particles with defects by 3d classification?

We increased the number of particles by ~ 4.7 times, and obtained similar results. Please see the new Supplementary Fig. 4b and c. Following this test, we updated the description in the text as,

“However, 2D class averages of ALSV images (Supplementary Fig. 4) and a 3D reconstruction assuming no icosahedral symmetry do not indicate such defects (Supplementary Fig. 4c). Here we picked up more particles from different datasets and processed together for this test (see Section 4.6 in Materials and Methods)”.

We also added Section 4.6 in Materials and Methods as,

“We picked up more ALSV particles from images taken with automated data collection software JADAS⁴⁷, although the yield rate was not good due to the uneven distribution of viruses in ice as above. Particle images were processed together with manually-collected ones, and 2D classification was applied to total 37,328 particles from 1,296 movie stacks with an option of “Ignore CTFs until first peak”, the setting of which is expected to be suitable for detection of finer features. A 3D map was reconstructed

without icosahedral symmetry form total 28,633 particles. Icosahedral symmetrisation of this combined dataset did not improve the resolution of the structure probably due to an increased heterogeneity among the datasets”.

> 7. The authors need to be sure to reference all relevant software used for image processing and model building/refinement. E.g. COOT for model building.

We used COOT to adjust the atomic model and superimpose the structures of ALSV and other picorna viruses, and MolProbity for refinement statistics. We also used DigitalMicrograph, SerialEM, ParalleEM and JADAS when image data were collected. We added these descriptions and the corresponding references.

> The conclusions made by the authors are reasonable given the data presented. However, a more thorough examination of genome release could be made with a larger dataset and further analysis. 3d classification of particles with/without RNA extruding and symmetry expansion and focussed classification of particles with/without extruding RNA would both likely give further insight into structural differences in at the vertices during RNA escape. Nonetheless the structure presented gives valuable insight into the structures of viruses within the cheravirus genus.

For reconstruction of the structure without the icosahedral symmetry, we increased the number of particles by ~ 4.7 times, and obtained similar results. Please see the new Supplementary Figs. 4b and c. In general, we can get a clear 2D average even from such an image number, as the signal-to-noise ratio is proportional to the square root of the image number (N). Indeed, we obtained clear 2D class averages from small numbers of particle images (Please see the new Supplementary Fig. 4b). For 2D classification of particle images with RNA leak, we also processed ~ 11 times more particle images and obtained similar class averages as in the new Fig. 3(c).

Again, it is difficult to obtain a 3D structure of this site, as RNA parts are random. Following the reviewer’s suggestion, we have tried a focused 3D reconstruction by masking one pentamer off but failed to obtain any solid structure even for that without RNA leak. This is because the signal-to-noise ratio is too low compared to the whole virion.

Finally, this is the first structure of the genus *Cheravirus* and the first cryo-EM structure newly determined with a cold-field emission electron source. We think more extensive analysis on RNA release is beyond the subject of this report and should be done in a separate study.

REVIEWERS' COMMENTS:

Reviewer #1 (Remarks to the Author):

The responses from the authors and the revised manuscript well addressed my questions or concerns. Given that this study resolved the high-resolution structure of the apple latent spherical virus (ALSV) and made some interesting discovery or observations about the mechanism of the viral capsid stabilization and genome release, I would like to recommend this manuscript to be published in Communications Biology.

Reviewer #3 (Remarks to the Author):

I thank the authors for their thorough response to my original comments, and can confirm that I am satisfied with the changes made to the manuscript. One small further comment is purely semantic.

> 3. The authors use 'similar protein folding' and to describe structures of the same protein fold. This may be confusing to the reader.

Corrected to "the same protein folding".

I would suggest changing this to 'the same protein fold'.